# Enhancement of Immune Response of Bioconjugate Nanovaccine by Loading of CpG through Click Chemistry

**DOI:** 10.3390/jpm13030507

**Published:** 2023-03-11

**Authors:** Mengting Mo, Xiang Li, Caixia Li, Kangfeng Wang, Shulei Li, Yan Guo, Peng Sun, Jun Wu, Ying Lu, Chao Pan, Hengliang Wang

**Affiliations:** 1College of Food Science and Technology, Shanghai Ocean University, No. 999 Hucheng Huan Road, LinGang New City, Shanghai 201306, China; 2State Key Laboratory of Pathogen and Biosecurity, Beijing Institute of Biotechnology, No. 20 Dongdajie Street, Fengtai District, Beijing 100071, China

**Keywords:** CpG, co-delivery, nanovaccine, click chemistry

## Abstract

CpG is a widely used adjuvant that enhances the cellular immune response by entering antigen-presenting cells and binding with receptors. The traditional physical mixing of the antigen and CpG adjuvant results in a low adjuvant utilization rate. Considering the efficient delivery capacity of nanovaccines, we developed an attractive strategy to covalently load CpG onto the nanovaccine, which realized the co-delivery of both CpG and the antigen. Briefly, the azide-modified CpG was conjugated to a bioconjugate nanovaccine (NP-OPS) against *Shigella flexneri* through a simple two-step reaction. After characterization of the novel vaccine (NP-OPS-CpG), a series of in vitro and in vivo experiments were performed, including in vivo imaging, lymph node sectioning, and dendritic cell stimulation, and the results showed that more CpG reached the lymph nodes after covalent coupling. Subsequent flow cytometry analysis of lymph nodes from immunized mice showed that the cellular immune response was greatly promoted by the nanovaccine coupled with CpG. Moreover, by analyzing the antibody subtypes of immunized mice, NP-OPS-CpG was found to further promote a Th1-biased immune response. Thus, we developed an attractive method to load CpG on a nanovaccine that is simple, convenient, and is especially suitable for immune enhancement of vaccines against intracellular bacteria.

## 1. Introduction

Vaccines are of great significance to human health, and currently, vaccination has become increasingly important for disease management given its wide applicability and long-term protection capability [1,2]. From the early inactivated or attenuated live vaccines to the current subunit vaccines, as well as the mRNA vaccines against COVID-19, many safer and more efficient vaccine systems are being continuously developed [3]. Subunit vaccines have weak immunogenicity that makes it difficult for them to stimulate an effective antibody response [4] and will need further modifications to overcome this bottleneck. In recent decades, many advanced delivery systems exhibiting both high efficacy and safety were developed. Antigens or epitopes can now be loaded efficiently in many ways to realize the targeted delivery. Nanovaccines, which have a pronounced ability to affect lymph node drainage and immune activation [2], have received widespread attention. In recent years, various nanoparticles, such as inorganic nanoparticles (NPs), inorganic and organic hybrid NPs, organic NPs, and proteinaceous NPs, have been used to develop bacterial vaccines and have shown powerful effects [2,5]. Prophylactic vaccines using proteinaceous nanomaterials (e.g., virus-like particles and ferritin) with a higher safety and biocompatibility are being widely studied [6]. With the development of synthetic biology, modular self-assembled nanoparticles are also being explored in vaccine design. In our previous study, we successfully prepared self-assembled nanocarriers by the fusion expression of pentamer and trimer domains and realized the loading of polysaccharide antigens in vivo to prepare a conjugate vaccine, which is now known as the most successful bacterial vaccine [7,8].

Although this bioconjugate nanovaccine strongly promotes an immune response without an adjuvant, for some intracellular bacteria (e.g., *Shigella flexneri* (*S. flexneri*) and *Brucella*), strong cellular immunity is more conducive to infection prevention [9,10]. Therefore, an appropriate adjuvant is necessary to further promote a cellular immune response. At present, the most commonly used aluminum adjuvants can only stimulate a humoral immune response, which is not effective against intracellular bacterial vaccines [11]. Many other efficient adjuvants have been developed that induce strong and specific immune responses, particularly cellular immune responses. For example, the pro-inflammatory cytokine interleukin 18 (IL-18) plays a key role in the induction of immune-mediated protection against intracellular pathogens [12,13]; the heat shock protein gp96 augments the antigen-specific cytotoxic T lymphocyte (CTL) response by binding and then internalizing into antigen-presenting cells (APCs) [14,15]; the Toll-like receptor 9 (TLR9) agonist CpG oligodeoxynucleotide (CpG ODN, also referred to as CpG) is a potent Th1 cell adjuvant that stimulates a strong B cell and natural killer cell activation [16,17]. CpG is a synthetic ODN containing an unmethylated cytosine guanine dinucleotide, which mimics bacterial DNA and stimulates immune cells in a variety of mammals, including humans [16,18,19]. As a component of a vaccine, it can enhance the activation of B cells, promote the maturation of plasmacytoid dendritic cells (pDCs), and induce the secretion of Th1-type cytokines [20,21]; moreover, it can increase vaccine-induced protective responses [22,23] and accelerate the development of vaccine-induced responses [24]. Notably, CpG is currently being evaluated in clinical trials as a potential vaccine adjuvant for COVID-19 vaccines [25]. Some studies also showed that CpG worked well with nanovaccines [21,26] and enhanced their therapeutic effect on tumors and immunostimulatory activity [27].

Traditionally, CpG is often coupled with vaccines through physical mixing [28]. For example, by mixing with CpG, pneumococcal conjugate vaccines significantly induced a cellular immune response [29]. However, this compatibility method requires a large amount of CpG, which increases the cost and potential safety risks. Considering that CpG needs to enter antigen-presenting cells to perform its biological function, researchers also tried to improve the utilization of CpG by coupling it with antigens [30,31]. At present, many CpG and antigen connectors, such as NHS-PEG_n_-maleimide (SMCC) and the His-hydrophobic-His (HUH) superfamily of endonucleases, have been explored [32,33,34]. Among them, SMCC, which contains an N-hydroxysuccinimide-active ester and maleimide, is used most frequently. However, SMCC is not suitable for antigens containing more than two cysteines [35].

Considering the importance of CpG in the cellular immune response, we attempted to realize the efficient utilization of CpG by its co-delivery with antigens through nano-delivery carriers, so as to stimulate a more efficient immune response. Since we have proved the outstanding delivery ability of the self-assembled nano-carriers for the OPS of *S. flexneri* 2a to further enhance the cellular immune response induced by the nanovaccine NP-OPS, we loaded CpG onto the NP-OPS by using a covalent connection. Briefly, CpG (modified by an azide group) was loaded onto the NP-OPS-containing alkyne group through the Cu(I)-catalyzed alkyne azide cycloaddition (CuAAC) reaction, which is a prime example of a current bio-orthogonal click chemistry reaction given its high reaction efficiency and mild reaction conditions [36,37]. After characterization of the novel nanovaccine, a series of in vitro and in vivo experiments indicated that, compared with physical mixing, the CpG covalent coupled to the nanovaccine was efficiently delivered to the lymph nodes and strongly promoted a cellular immune response. Moreover, by analyzing the antibody subtypes of immunized mice, NP-OPS-CpG was found to further promote a Th1-biased immune response. Therefore, we provided a novel and attractive method to load CpG on a nanovaccine to further improve the cellular immune response. It is worth emphasizing that this method is simple, convenient, and is especially suitable for the immune enhancement of intracellular bacterial vaccines.

## 2. Materials and Methods

### 2.1. Bacterial Strains and Growth Conditions

NP-OPS was expressed in *Shigella* 301DWP containing the pET28a-pglL-CTBtri4573 plasmid as previously described [7]. Bacteria were cultured in Luria–Bertani (LB) medium at 37 °C (220 rpm). For expression, cells were cultured to an optical density value of 0.6–0.8 at 600 nm and induced with a final concentration of 1 mM isopropyl-beta-D-thiogalactopyranoside (IPTG) to express NP-OPS. The culture then continued to incubate at 30 °C for 14 h (220 rpm).

### 2.2. Preparation of NP-OPS-CpG via Click Chemistry

Based on the principle of the covalent binding of free amino groups in proteins to succinimide, we mixed NP-OPS with succinimide-PEG_4_-alkyne at 25 °C for 2 h. Then, the mixture was dialyzed in phosphate-buffered saline (PBS) at 4 °C for 2 days. The PBS was replaced every 6 h with a final change using ddH_2_O to obtain pure NP-OPS-alkyne. CpG and its modified products were purchased from Generay. Based on the principle of the Cu-catalyzed azide-alkyne cycloaddition reaction, NP-OPS-CpG was obtained by combining lyophilized NP-OPS-alkyne with azide-modified CpG2006 (N_3_-CpG, CpG2006: 5′-TCGTCGTTTTTGTCGTTGTCGTT-3′) by using a Click-iT protein buffer kit (Thermo Fisher, Waltham, MA, USA). Finally, the purified NP-OPS-CpG was obtained using size-exclusion chromatography through a Superdex-200 column (GE Healthcare, Chicago, IL, USA) in a mobile phase consisting of PBS, as described previously.

### 2.3. Coomassie Blue Staining and Western Blotting

The preparation of protein samples was carried out as described previously with slight modifications [7]. The purified NP-OPS, NP-OPS-alkyne, and NP-OPS-CpG were verified with the Coomassie blue staining method. The glycoprotein samples were detected through Western blotting with anti-6 × His antibody (Abmart, Shanghai, China) and anti-*S. flexneri* OPS-specific serum (Denka Seiken, Tokyo, Japan).

### 2.4. Stimulation of DC2.4 Mouse Dendritic Cell Line (DC2.4s) by Vaccines In Vitro

The nucleic acid concentration of the sample NP-OPS-CpG was measured by NanoDrop^TM^ 2000 (Thermo Fisher, Waltham, MA, USA). CpG was added to the NP-OPS (without alkyne) to obtain the sample of NP-OPS+CpG. To ensure the same amount of nucleic acid in treat, the CpG concentration in NP-OPS+CpG sample was measured to be consistent with that of NP-OPS-CpG. Untreated DC2.4s were cultured to 100,000 cells/well at 37 °C with 5% CO_2_. DC2.4s were stimulated with PBS, CpG_Cy5_, NP-OPS mixed with CpG_Cy5_ (NP-OPS+CpG_Cy5_), and NP-OPS-coupled CpG_Cy5_ (NP-OPS-CpG_Cy5_). Each treat contained 50 ng CpG_Cy5_. After incubating for 6 h, 12 h, or 24 h, DC2.4s were digested with trypsin, and free cells were obtained by centrifugation at 4 °C (500× *g*, 5 min). Cells were centrifuged and resuspended in 100 μL of cold staining buffer (eBioscience, San Diego, CA, USA), and this step was repeated twice. The washed cells were filtered through a 200-mesh screen to obtain samples for flow cytometry. Finally, Cy5-labeled DC2.4s were analyzed using a CytoFLEX LX flow cytometer (Beckman, Brea, CA, USA).

### 2.5. Lymph Node Imaging Assay

Mice were randomly divided into three groups (CpG_Cy7_, NP-OPS+CpG_Cy7_, and NP-OPS-CpG_Cy7_). Mice were injected in the tail base with samples that had a consistent fluorescence intensity. Mouse lymph node fluorescence signals at different time points (0 h, 6 h, 12 h, and 24 h after injection) were measured by an IVIS spectrum in vivo imaging system (PerkinElmer, Waltham, MA, USA).

### 2.6. Mouse Immunization

Isonucleic acid concentrations of NP-OPS-CPG and NP-OPS+CpG were prepared as described above. Specific-pathogen-free female BALB/c mice (6–8 weeks old) were purchased from Spaifer and were randomly divided into 4 groups (10 mice in each group). Mice were subcutaneously injected with 100 μL of PBS, NP-OPS, NP-OPS+CpG, or NP-OPS-CpG with a polysaccharide dose of 2.5 μg/mouse on days 0, 14, and 28. Corresponding CpG in both NP-OPS+CpG and NP-OPS-CpG groups was 120 ng/mouse. One week after the last immunization, blood samples were taken by tail clipping, and serum was stored at 4 °C. All animal experiments were approved and conducted by the institutional guidelines of the Academy of Military Medical Sciences and with the approval of the Institutional Animal Care and Use Committee (Approval Code IACUC-DWZX-2021-073).

### 2.7. T Cell Immune Response Induced by the Vaccines In Vivo

Mice were grouped and immunized as described previously with slight modifications [7]. BALB/c mice were immunized with one of three formulations (NP-OPS, NP-OPS+CpG, and NP-OPS-CpG), and draining lymph nodes (dLNs) of each mouse were individually collected three days post-vaccination and triturated into single cell suspension. Then, the cells were stained with APC-conjugated anti-mouse CD3. After fixation and permeabilization, the cells were further stained with FITC-conjugated anti-mouse Ki 67 antibody and analyzed by flow cytometry.

Five days after the third immunization, the mice were sacrificed to obtain lymph nodes. Flow cytometry samples were prepared as described above. Cells were stained with the following antibodies: APC-conjugated anti-mouse CD3, FITC-conjugated anti-mouse CD4, and PE-conjugated anti-mouse CD8. CD3^+^, CD4^+^, and CD8^+^ cells were analyzed by a CytoFLEX LX flow cytometer (Beckman, Brea, CA, USA). All flow cytometry antibodies were purchased from eBioscience, San Diego, CA, USA.

### 2.8. ELISA

*S. flexneri* 2a lipopolysaccharide (LPS) (100 μg/well) was diluted with a coating solution and plated onto 96-well plates that were incubated overnight at 4 °C. The plates were washed 3 times with PBST (PBS with 0.05% Tween) and blocked with 5% skim milk at 37 °C for 2 h. After washing the plates 3 times, serum diluted with a holding solution in a 2-fold serial ratio was added to the plates and incubated at 37 °C for 1 h. After again washing 3 times, 100 μL of HRP-conjugated goat anti-mouse IgG (Abcam, Cambridge, UK) (1:15,000) was added and incubated at 37 °C for 1 h. After washing the plate again, 100 μL of tetramethylbenzidine solution (CWBio, Beijing, China) was added for the color reaction. Finally, the absorbance value of each well at 450 nm was measured after adding the termination solution.

### 2.9. Immune Effects of the NP-OPS-CpG in an S. flexneri 2a Infection Model

Next, 14 days after 3 immunizations, mice were challenged with 2.5 × 10^7^ CFU *S. flexneri* 2a by intraperitoneal injection (i.p.), and the survival of each group of mice was monitored continuously for 14 days (n = 10).

### 2.10. Statistical Analysis

All data in these experiments were analyzed by GraphPad Prism 7.0 statistical software (GraphPad Inc, San Diego, CA, USA). The data were analyzed using one-way ANOVA and *t*-test. Results were expressed as means ± SDs. Values of *p* < 0.05 were considered statistically significant (**** *p* < 0.0001, *** *p* < 0.001, ** *p* < 0.01, and * *p* < 0.05).

## 3. Results

### 3.1. Conjugation of NP-OPS and CpG

The NP-OPS-CpG conjugate vaccines were prepared as described in the experimental protocol (Figure 1a). First, to modify the alkyne group on NP-OPS, NP-OPS was incubated with succinimide-PEG_4_-alkyne at 25 °C for 2 h. After removing unbound succinimide-PEG_4_-alkyne through dialysis, NP-OPS containing the alkyne group was further incubated with N_3_-CpG at 25 °C for 20 min. Then, NP-OPS-CpG was obtained by size-exclusion chromatography. After separating samples from each step in the process using SDS-PAGE, results from the Coomassie blue staining and Western blot with an anti-6 × His tag antibody and anti-*S. flexneri* OPS serum showed that the typical NP-OPS ladder had an obvious upward migration after each step, indicating the successful loading of succinimide-PEG_4_-alkyne and CpG separately (Figure 1b). As expected, the size-exclusion chromatography results showed that the retention volume of NP-OPS-CpG was 8–11 mL, about 30% of the column volume, indicating that NP-OPS-CpG was still in the form of a polymer (Figure 1c). To further determine the combination, different concentrations of FAM-labeled N_3_-CpG were incubated with a certain amount of NP-OPS-alkyne, as described above. After separating by SDS-PAGE, samples were analyzed by Coomassie blue staining and observed under ultraviolet light. The results showed that with an increased concentration of CpG, the fluorescence intensity of the NP-OPS band increased, suggesting that CpG coupled with NP-OPS (Figure 1d).

### 3.2. Analysis of the NP-OPS-CpG Bioconjugate Nanovaccine

Previous results showed that NP-OPS existed in the form of nanoparticles [7]. To determine whether the coupling of CpG affected the formation of particles, purified NP-OPS-CpG was analyzed using transmission electron microscopy (TEM). The results showed that NP-OPS-CpG was about 20–30 nm in diameter (Figure 2a). Dynamic light scattering (DLS) results also revealed that NP-OPS-CpG was monodisperse and was about 29.76 nm, which was in line with the TEM result and consistent with that before coupling due to the small molecular weight of CpG. The polydispersity index (PDI) was about 0.2698 (slightly higher than 0.2), which may be attributed to the nonuniform length of polysaccharide antigens on the nanoparticle. Although the NP-OPS was almost uncharged due to the coverage of polysaccharides which had no charge itself, the coupling of CpG made it carry negative charges, also suggesting that CpG was successfully coupled to the surface of NP-OPS (Appendix A). In addition, by measuring the concentrations of nucleic acid and proteins in NP-OPS-CpG, respectively, we calculated that there were about 40 CpG on each particle. Then, we analyzed the stability of NP-OPS-CpG and found that it maintained stability after being stored at 37 °C for at least 7 days (Figure 2c).

### 3.3. Evaluation of CpG Lymph Node Targeting

One of the advantages of nanovaccines is efficient lymph node drainage, enabling more antigens to reach immune organs. Because the CpG in our design was covalently coupled to the bioconjugate nanovaccine, it would also be delivered, in theory. To confirm this, Cy5-labeled CpG was coupled with NP-OPS, and the three treatments (CpG_Cy5_, NP-OPS+CpG_Cy5_, or NP-OPS-CpG_Cy5_) were injected into the tail base of BALB/c mice. DLNs were taken from each mouse 6 h post-injection, and the fluorescence imaging results of dLNs sections showed that stronger CpG signals were detected in NP-OPS-CpG-immunized mice compared to the other mice (Figure 3a). Then, Cy7-labeled CpG was coupled to NP-OPS, and mice were injected as above. In vivo imaging results showed that the intensity of the signal for CpG alone at the site of the dLNs was barely detected at any time point. The signal for NP-OPS+CpG_Cy7_ revealed an obvious accumulation pattern in the dLNs, possibly due to a small amount of non-specific binding. In contrast, dramatic increases in lymph node accumulation were observed for NP-OPS-CpG_Cy7_, which was attributed to the co-delivery with the nanovaccine (Figure 3b). Furthermore, the analysis of the total signal intensity throughout the time course experiment revealed an 11 times increase in the lymph-node-specific accumulation of NP-OPS-CpG over CpG (Figure 3c).

### 3.4. Analysis of CpG Phagocytosis by DC2.4s

Having confirmed the efficient lymph node drainage of CpG loaded onto the nanovaccine, we further analyzed the endocytic activity of DCs in different forms of CpG. DC2.4 is a mouse bone marrow dendritic cell line established with C57BL/6 mouse bone marrow cells transfected by the V-myc and V-raf genes and used to simulate the function of APC in several studies [30,38]. By stimulating DC2.4s with CpG_Cy5_, NP-OPS+CpG_Cy5_, or NP-OPS-CpG_Cy5_, the amount of CpG phagocytosed by DC2.4s was analyzed by flow cytometry. The results showed that although Cy5^+^ in the DC2.4s in the CpG_Cy5_ and NP-OPS+CpG_Cy5_ treatment groups increased at the three time points, the greatest Cy5 signals were detected in the NP-OPS-CpG_Cy5_ group (Figure 4a,b). Particularly when incubated for 24 h, the phagocytic efficiency was about four times higher than that of CpG alone (Appendix A).

### 3.5. T Cell Proliferation and Differentiation Induced by the Nanovaccines

To explore the effects of the different vaccines on the proliferation and differentiation of T cells, BALB/c mice were immunized with NP-OPS, NP-OPS+CpG, or NP-OPS-CpG. Three days after immunization, mice were sacrificed, the dLNs were removed, and Ki67^+^ and CD3^+^ cells in dLNs were analyzed by flow cytometry. The results showed the percentage of Ki67-positive T cells was significantly increased in the NP-OPS-CpG-treated group compared with that of other groups (Figure 5a). In addition, after three immunizations at two-week intervals, mice were sacrificed five days after the last injections, and dLNs were removed. Flow cytometry results showed that NP-OPS-CpG-injected mice had a significant increase in the ratio of CD3^+^ cells (Appendix A). The proportion of CD4^+^ T cells and CD8^+^ T cells in lymph nodes was also analyzed, and the results showed that the NP-OPS-CpG-injected mice had the greatest increase in both subtypes (Figure 5b,c). In particular, CD8^+^ levels in the NP-OPS-CpG group increased more than in the NP-OPS+CpG group (Figure 5c).

### 3.6. Antibody Response and Protective Effect in NP-OPS-CpG Immunized Mice

To evaluate the antibody response induced by NP-OPS-CpG, BALB/c mice were immunized with 1 of 4 treatments (PBS control, NP-OPS, NP-OPS+CpG, or NP-OPS-CpG) on days 0, 14, and 28. Blood was sampled on day 38 to facilitate the quantitation of antibodies against *S. flexneri* 2a LPS. The bacterial pathogen challenge was administered on day 42, followed by the monitoring of mouse survival (Figure 6a). ELISA results revealed that NP-OPS-CpG induced a higher IgG titer, although it had no statistical significance (Figure 6b). Subsequently, titers of the IgG subtype (IgG1 and IgG2a) were measured, and the results showed that the NP-OPS-CpG-treated group induced significantly higher IgG2a titers, suggesting further promotion of a Th1-biased immune response (Figure 6c). In addition, by calculating the ratio of IgG1 and IgG2a of each group, it was found that NP-OPS coupled with CpG revealed a significantly lower ratio than that of the NP-OPS+CpG group (Figure 6c). These results indicated that although physical mixing of CpG improved the Th1 immune response, coupling was more conducive to establishing a balance favoring a Th1-biased immune response. Then, mice were injected intraperitoneally with a dose of 2.5 × 10^7^ CFU per mouse of *S. flexneri* 2a strain 14 days after the third immunization, and the survival of each group of mice was observed. All the mice in the PBS group died rapidly in the first two days, and all mice in the other three groups were alive (Figure 6d). The results indicated that by coupling with CpG, NP-OPS maintained efficient prophylactic effects against infection.

## 4. Discussion

In this study, we developed an attractive strategy for producing a bioconjugate nanovaccine loaded with a CpG adjuvant. Different from traditional physical mixing, the covalent coupling of CpG and NP-OPS realized the co-delivery of an antigen and CpG through nano-carriers. CpG was rapidly drained to the lymph nodes by using nano-carriers and was easily engulfed by antigen-presenting cells. Subsequently, the cellular immune response was greatly enhanced. Therefore, we provided here a novel method for loading CpG onto a nanovaccine. This method is simple, convenient, and is especially suitable for the immune enhancement of intracellular bacterial vaccines.

In recent years, the CpG adjuvant has been widely used in vaccine research. Particularly for COVID-19 vaccines, CpG is often used together with a classical aluminum adjuvant [39]. By adsorbing to an aluminum salt through electrostatic action, CpG utilized the storage effect of the aluminum adjuvant and was beneficial for activating a cellular immune response. However, this compatibility may not be suitable for nanovaccines. Generally, one of the advantages of nanovaccines is the size-related ability to rapidly drain to and accumulate in lymph nodes [40]. As previously reported, 15–100 nm is the optimal size of vaccines for direct homing to draining lymph nodes [41]. Thus, this advantage is weakened if an aluminum adjuvant is added. Furthermore, our previous study also showed that the addition of an aluminum adjuvant to the bioconjugate nanovaccine did not further improve the antibody response [8]. Therefore, coupling with CpG is appropriate for improving the bioconjugate nanovaccine response. This conjugation not only maintained the size advantage of the vaccine, but also realized the co-delivery of CpG and the antigen. In our results, more CpG, when coupled with NP-OPS, reached the lymph nodes with a better uptake by DCs compared to a physical mixture with CpG; thus, the coupling strategy was more efficient and conducive to stimulating a cellular immune response.

In our study, CpG was covalently coupled with the nanovaccine through a succinimide-PEG_4_-alkyne-bridge. Many other coupling modes have been developed. At present, the most widely used coupling agent is SMCC, which contains N-hydroxysuccinimide active ester and maleimide. The two active groups of SMCC couple CpG with a protein antigen. However, because disulfide bonds need to be opened during the crosslinking process, SMCC is not suitable for antigens containing more than two cysteines [35]. Our study selected succinimide-PEG_4_-alkyne as a connector for which active groups can complete the reaction quickly under mild conditions [42]. At the same time, no influence on the protein structure was found. Therefore, it was suitable for proteinaceous nano-carriers. In addition, the protein HUH also binds to CpG. By fusion expression with protein antigens, HUH can serve as a bridge to load CpG. Considering the influence of HUH expression on nanostructures, this strategy is more suitable for monomer protein carriers.

Our results have shown that the covalent coupling of CpG and the delivery vector was more efficient than the physical mixing to induce a cellular immune response. However, there still were some possible limitations in this study. For example, to better evaluate the effect of the NP-OPS-CpG conjugate vaccine, a group treated with a traditional adjuvant (such as aluminum) could be added as a control to determine the relative efficacy. In addition, in our research, we focused on the vaccine’s efficacy against the *S. flexneri* infection. Considering that the active groups we selected can complete the reaction quickly under mild conditions, universal applicability could be explored, and the immune enhancement effect of serious vaccines against other pathogens (especially intracellular bacteria) can also be further measured. Moreover, increasing the sample size in future studies would strengthen the findings. In addition, because this coupling method increases the CpG utilization rate, the dosage of CpG can be reduced and compared with the current dosage, thus reducing the production cost and the side effects. This method also provides direction for the research of a new generation of self-adjuvant vaccines. Moreover, through modifications of nanostructures and amino acids in the future, more binding sites can be designed to realize controllable CpG loading.

## Figures and Tables

**Figure 1 jpm-13-00507-f001:**
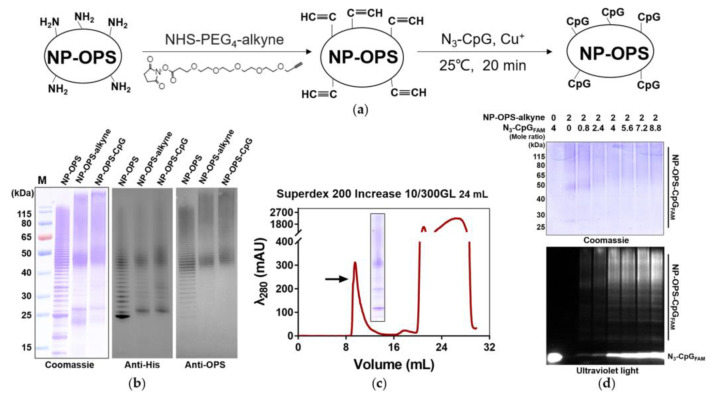
Preparation of the NP-OPS-CpG conjugate vaccine. (**a**) Schematic diagram of the NP-OPS-CpG preparation. (**b**) Purified NP-OPS, NP-OPS-alkyne, and NP-OPS-CpG were detected by Coomassie blue staining and Western blotting with antibodies against the 6 × His tag and *S. flexneri* OPS. (**c**) Analysis of purified NP-OPS-CpG by size-exclusion chromatography (Superdex 200, 24 mL). The peak position of NP-OPS-CpG is indicated by an arrow, and detected by Coomassie blue staining. (**d**) NP-OPS-CpG_FAM_ obtained with different NP-OPS-alkyne and N_3_-CpG_FAM_ ratios was separated by SDS-PAGE, analyzed by Coomassie blue staining, and observed under ultraviolet light.

**Figure 2 jpm-13-00507-f002:**
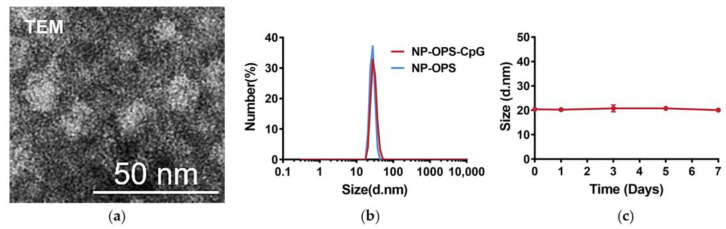
Characterization and stability of NP-OPS-CpG. (**a**) TEM images of NP-OPS-CpG. (**b**) DLS analysis of NP-OPS and NP-OPS-CpG separately, showing that the nanoparticles were about 25–50 nm in diameter. (**c**) Stability of the NP-OPS-CpG diameter after being stored at 37 °C for 7 days. The diameter of NP-OPS-CpG was analyzed by DLS (n = 3).

**Figure 3 jpm-13-00507-f003:**
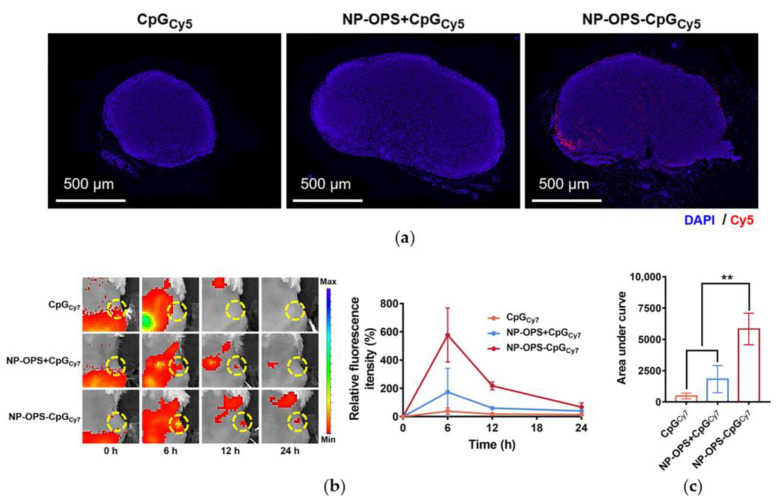
Lymph node targeting of the bioconjugate nanovaccine. (**a**) NP-OPS-CpG_Cy5_ was injected into the tail base of mice with CpG_Cy5_ and NP-OPS+CpG_Cy5_ as controls. Lymph nodes were obtained from 6 h post-injection mice. Cryosections of dLNs were stained with DAPI (DAPI: blue; Cy5: red) (n = 3). (**b**) Representative images and corresponding quantitative fluorescence analyses of different vaccines (CpG labeled with Cy7) in dLNs (n = 3). The dLNs site was circled as shown. (**c**) Accumulation of CpG (labeled by Cy7) in lymph nodes within 24 h (n = 3). Data are presented as means ± SD. Each group was compared with NP-OPS-CpG_Cy7_ using one-way ANOVA: ** *p* < 0.01.

**Figure 4 jpm-13-00507-f004:**
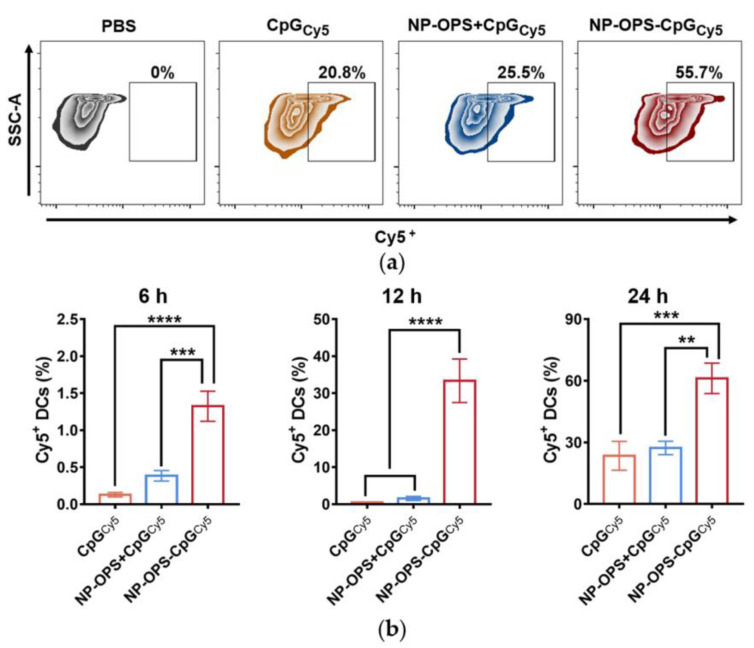
DC2.4s phagocytosis of the nanovaccine. (**a**) Representative flow cytometry gating for evaluating Cy5^+^ production in DC2.4s. (**b**) DC2.4s phagocytosis of the bioconjugate nanovaccine 6 h, 12 h, and 24 h after being stimulated by CpG_Cy5_, NP-OPS+CpG_Cy5_, and NP-OPS-CpG_Cy5_ separately (n = 3). Data are presented as means ± SD. Each group was compared with NP-OPS-CpG_Cy5_ using one-way ANOVA: **** *p* < 0.0001, *** *p* < 0.001, and ** *p* < 0.01.

**Figure 5 jpm-13-00507-f005:**
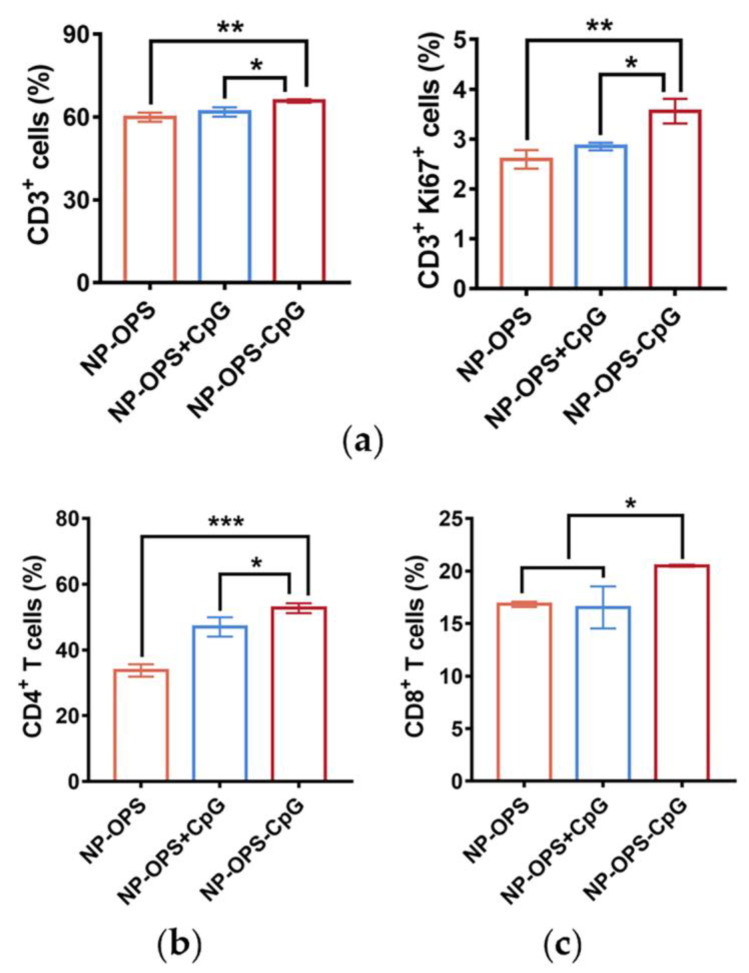
T cell immune responses induced by NP-OPS-CpG in vivo. (**a**) NP-OPS, NP-OPS+CpG, and NP-OPS-CpG were injected into the tail base of mice separately (n = 3). Lymph nodes were obtained from mice three days after immunization, and CD3^+^ and CD3^+^ Ki67^+^ cells were analyzed by flow cytometry. (**b**,**c**) NP-OPS, NP-OPS+CpG, and NP-OPS-CpG were injected into the tail base of mice separately (n = 3). Lymph nodes were obtained from mice five days after the third immunization, and CD4^+^ (**b**) and CD8^+^ (**c**) cells were analyzed by flow cytometry (n = 3). Each group was compared with NP-OPS-CpG using one-way ANOVA: *** *p* < 0.001; ** *p* < 0.001, and * *p* < 0.05.

**Figure 6 jpm-13-00507-f006:**
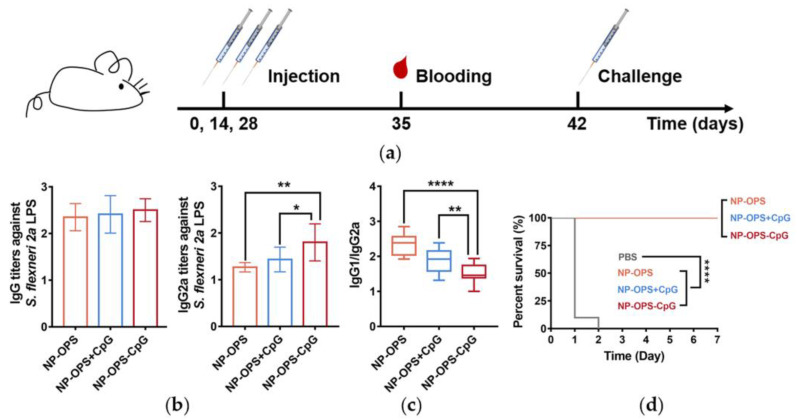
Protection induced by the nanovaccines. (**a**) Immunization schedule for the titer evaluation. (**b**) Total IgG and IgG2a titers against *S. flexneri* 2a LPS in immunized mice were measured after the third immunization (n = 10). (**c**) The ratio of IgG1 and IgG2a titers was calculated. (**d**) Immunized mice were injected i.p. with the *S. flexneri* 2a strain (2.5 × 10^7^ CFU per mouse) seven days after the final immunization, and their survival was monitored (n = 10). Data are represented as mean ± SD. Each group was compared with NP-OPS-CpG using one-way ANOVA: **** *p* < 0.0001; ** *p* < 0.01; and * *p* < 0.05.

## Data Availability

The data presented in this study are available on request from the corresponding author.

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
