# Peer review of "Enhancement of Immune Response of Bioconjugate Nanovaccine by Loading of CpG through Click Chemistry"

_jpm, 2023, doi:10.3390/jpm13030507_

Round 1
Reviewer 1 Report
The authors present a nice study that builds on their previous works on the development of a nanovaccine platform. Here, they explore a well-known alkyne-azide reaction to couple their platform with CpG adjuvant, and perform the relevant tests in vivo and on dendritic cells. Overall, the rationale and methods are well explained and the experiments support the conclusions. However, a few points must be addressed before I can suggest the publication on JPM.
1. Please define acronym when they first appear in the text (e.g. SMCC, HUH, DC2.4…).
2. Characterization of NP-OPS-CpG. I suggest to improve it addressing the following points:
2a. Include the polydispersity index as a measure of particle size distribution.
2b. Define the number of CpGs bound to each NP-OPS (mol/mol ratio or alternative measure). This is a critical characterization to allow reproducibility of the production and better evaluate the impact of immunological results.
3. CpG phagocytosis by DC2.4s. Please specify the dose administered to cells. This is connected to the previous point: how did you ensure that the same CpG dose was used on cells if the number of CpGs on NP-OPS was not quantified (or at least not reported in the manuscript)?
4. Mouse immunization. Same point as before: it is not clear from the text if the dose of CpG in the two different forms (free VS conjugated) was the same for the three groups. Please specify in the methods section how the dose was measured, or discuss this point as a limit of the study.
Reviewer 2 Report
Dear Authors,
I have had the opportunity to review your manuscript titled "Enhancement of Immune Response of Bioconjugate Nanovaccine by Loading of CpG through Click Chemistry". Overall, the study presents a novel method for loading CpG onto a nanovaccine, which has potential applications for improving immune responses to intracellular bacterial vaccines. I have reviewed your manuscript, and I would like to provide some comments to help improve the quality of your study.
Firstly, the introduction provides a clear and comprehensive overview of the topic of the study, including the background and significance of the research. However, it lacks a clear and concise statement of the objectives or aims of the study, which would help to guide the reader through the research. I suggest adding a brief section at the end of the introduction that outlines the study's specific aims and objectives.
Secondly, while the methods section provides a detailed description of the experimental procedures, some important details are missing. For example, the authors do not describe the specific CpG oligonucleotide sequence used, which is important for replication and comparison by other researchers. In addition, the authors do not provide a complete characterization of the nanoparticle's physical properties, such as its size, charge, and surface composition. These details are important to understand the behavior of the nanoparticle in biological systems and to ensure reproducibility.
Thirdly, while the results provide strong evidence that the NP-OPS-CpG conjugate vaccine promotes T cell expansions and antibody production, the study does not demonstrate proliferation explicitly. The authors could improve the study by including assays for the proliferation marker Ki67 in their T cell analyses.
Finally, the conclusion provides a clear summary of the results but lacks a critical analysis of the limitations and implications of the study. The authors could improve the conclusion by discussing the limitations of the study, such as the lack of comparison groups, the study did not include comparison groups, such as a group treated with a traditional adjuvant or a group treated with an aluminum adjuvant, which makes it difficult to determine the relative efficacy of the NP-OPS-CpG conjugate vaccine, and the limited scope, the study only evaluated the vaccine's efficacy against S. flexneri infection, which may limit its applicability to other bacterial infections. Additionally, the sample size in some experiments is relatively small (n=3), which may limit the statistical power of their results. Increasing the sample size in future studies would strengthen their findings.
Overall, I believe that the manuscript has the potential to make a valuable contribution to the field of vaccine development. I recommend that the authors revise their manuscript to address the aforementioned concerns before it can be considered for publication.
Round 2
Reviewer 1 Report
The authors have addressed all the comments; I can recommend this manuscript for publication in JPM.